# Autocatalytic base editing for RNA-responsive translational control

**Raphaël V. Gayet** [1,2,3,4,7], **Katherine Ilia** [1,2,7], **Shiva Razavi** [1,2,4,5,7], **Nathaniel D. Tippens**[1,2,4,7], **Makoto A. Lalwani**[2,4], **Kehan Zhang** [2,4], **Jack X. Chen**[1,2,4], **Jonathan C. Chen**[1,2,5], **Jose Vargas-Asencio** [6] & **James J. Collins** [1,2,4,5] ✉

Genetic circuits that control transgene expression in response to pre-defined transcriptional cues would enable the development of smart therapeutics. To this end, here we engineer programmable single-transcript RNA sensors in which adenosine deaminases acting on RNA (ADARs) autocatalytically convert target hybridization into a translational output. Dubbed DART VADAR (Detection and Amplification of RNA Triggers via ADAR), our system amplifies the signal from editing by endogenous ADAR through a positive feedback loop. Amplification is mediated by the expression of a hyperactive, minimal ADAR variant and its recruitment to the edit site via an orthogonal RNA targeting mechanism. This topology confers high dynamic range, low background, minimal off-target effects, and a small genetic footprint. We leverage DART VADAR to detect single nucleotide polymorphisms and modulate translation in response to endogenous transcript levels in mammalian cells.

Developing robust tools to modulate the activity of genetic payloads in response to pre-defined cellular cues is a pressing challenge in biomedicine and biological engineering[1]. Context-aware genetic circuits would have extensive applications in clinical settings as they could adjust gene expression during disease progression or facilitate precise, cell-specific targeting while minimizing off-target effects[2]. Notably, recent advances in transcriptomics have generated rich datasets that capture the molecular signatures of cell states and cell types[3], motivating efforts to harness this information for the selective, on-demand expression of therapeutic transgenes using novel sense-and-respond modules.

Transcripts of interest can be detected upon specific hybridization with engineered RNA molecules. As such, a key advantage of RNA as a sensor module is its ability to detect targets of interest by simple base pairing, thus facilitating the design of highly programmable tools that can be easily repurposed for new applications[4,5]. In particular, strand displacement has been explored as a strategy for the direct sensing of RNA triggers both in prokaryotes and eukaryotes[6,7].

Recently, the repertoire of transcript-sensing riboregulators was broadened to eukaryotes in a technology known as eToeholds, which relies on engineered mRNA internal ribosome entry sites (IRES)[8]. In eToeholds, inhibitory loops of IRES structures are disrupted upon hybridization with target RNAs, thereby restoring ribosome recruitment and enabling RNA-responsive translational control.

Most recently, three groups have independently described convergent approaches to design RNA-based sensors[9–11]. In these preliminary reports, base editing by adenosine deaminases acting on RNAs (ADARs) couples the detection of an RNA trigger to the translation of a user-defined genetic payload (Fig. 1a). ADARs efficiently edit mismatched adenosines within imperfect double-stranded RNA (dsRNA) structures[12]. The specific hybridization of an engineered sensor transcript with an RNA target of interest therefore allows the conditional recruitment of these RNA-editing enzymes to pre-defined edit sites on the sensors. As adenosines and inosines are interpreted differently by the translational machinery[13], sensor transcript sequences can be designed such that an in-frame UAG stop codon is

[1]Department of Biological Engineering, Massachusetts Institute of Technology (MIT), Cambridge, MA, USA. [2]Institute for Medical Engineering and Science, MIT, Cambridge, MA, USA. [3]Microbiology Graduate Program, MIT, Cambridge, MA, USA. [4]Wyss Institute for Biologically Inspired Engineering, Harvard University, Boston, MA, USA. [5]Broad Institute of MIT and Harvard, Cambridge, MA, USA. [6]Picower Institute for Learning and Memory, MIT, Cambridge, MA, USA. [7]These authors contributed equally: Raphaël V. Gayet, Katherine Ilia, Shiva Razavi, Nathaniel D. Tippens. ✉e-mail: jimjc@mit.edu

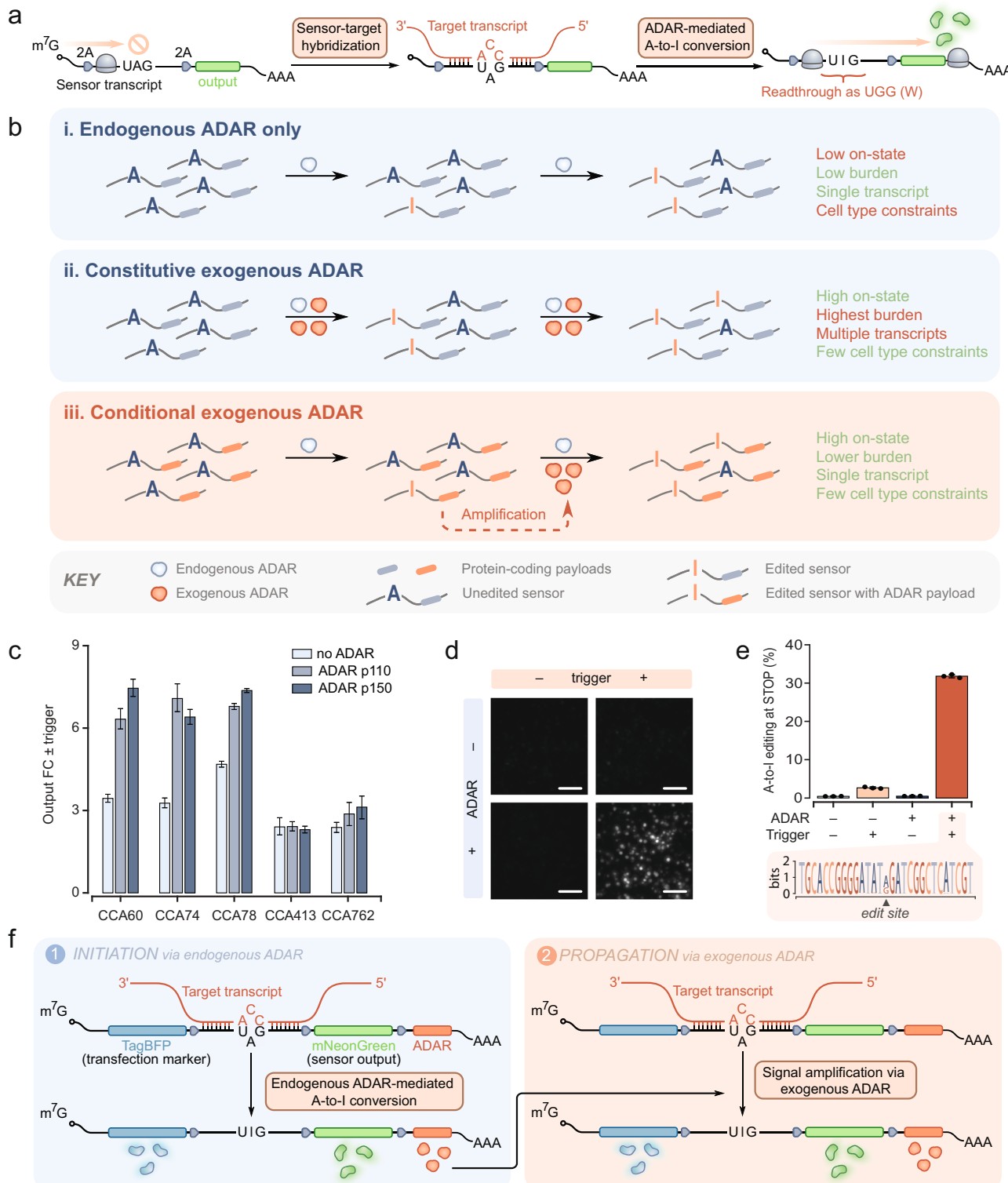

converted to UIG. As a result of the base editing, the amber codon becomes a sense (tryptophan) codon that does not block translation, leading to the expression of a protein payload encoded downstream of the edited codon (Fig. 1a). Through this process, ADAR enzymes convert the detection of an RNA target (via base pairing) into translational activation.

As ADARs are ubiquitous in metazoan cells, these sensors could be used in isolation to detect RNA molecules of interest (Fig. 1b). Although this design paradigm has been validated in vivo in neurons[10], the nervous system is known to express high levels of

ADAR[14]. Therefore, circuits using endogenous levels of ADARs might not be as effective in other tissues. Overexpression of exogenous ADAR has been explored as a possible solution to enhance the performance of this class of circuits in cells with a lower supply of endogenous ADAR[9,11] (Fig. 1b). This, however, results in an increase in the number of transcriptional units, which may hinder the delivery of such a system to cells of interest. In addition, wild-type ADAR enzymes are promiscuous, and their overexpression may lead to off-target effects. Together, these limitations highlight the need for compact, modular, and self-sufficient circuit designs

**Fig. 1 | Autocatalytic DART VADAR sensors are a practical implementation of ADAR-mediated RNA-responsive translational control. a** Schematic presenting an overview of a basic ADAR-mediated RNA-responsive translational switch. These sensors are activated by the specific hybridization of target transcripts, followed by the enzymatic deamination of the mismatched A in the central stop codon.
**b** Illustrated are three different ways to design such circuits; these, as well as pros (in green) and cons (in orange) inherent in these designs, are summarized in panels i, ii, and iii. (i) These sensors can be designed such that there is no exogenous supplementation of ADAR[10]. In such systems, the low levels of endogenous ADAR carry out editing of a subset of sensor molecules. (ii) Other implementations rely on constitutive overexpression of exogenous ADAR[9,11], which efficiently mediates editing of sensor molecules while sacrificing ease-of-delivery and increasing the unnecessary consumption of cellular resources. (iii) A potential solution that builds on these approaches is based on conditional expression of exogenous ADAR. Here, endogenous ADAR mediates editing in a subset of sensor molecules, prompting the translation of the circuit payload, including ADAR itself. After this initial step,

exogenously produced ADAR increases the frequency of editing events and consequently yields higher dynamic range. m7G: mRNA cap; 2A: self-cleaving 2A peptide; AAA: poly(A) tail. **c** Exogenous supplementation of ADAR improves sensor performance. Numbers following the CCAs indicate the nucleotide position of the central target triplet, using the start codon as position +1. The value of each bar corresponds to the output fold-change (FC), which is the ratio of the geometric mean of mNeonGreen expression in the presence and absence of trigger. Error bars represent 95% confidence intervals for the fold-change values, determined from at least 2000 cells. **d** Fluorescence microscopy of mNeonGreen illustrates CCA60 sensor performance against iRFP720 in HEK293FT cells, 48 hr after transfection (Scale bar: 300 μm). **e** Sequencing data confirms increased A-to-I editing of the CCA60 sensor in the presence of trigger and exogenous ADAR p150. Error bars correspond to the standard deviation for $n = 3$ biological replicates. The sequence logo demonstrates ADAR-mediated editing is specific to the central A in the UAG stop codon. **f** DART VADAR relies on an initial editing step by endogenous ADAR, which is then amplified by exogenous ADAR.

that operate across various cell types irrespective of the cell's resource context.

Here, we describe a circuit topology that overcomes the constraints imposed by the endogenous ADAR supply without stymying delivery. We hypothesized that an autocatalytic circuit activated by ADAR expressed at endogenous levels could be engineered to express not only the desired protein output of interest, but also ADAR itself to edit additional sensors (Fig. 1b). This led us to design DART VADAR (Detection and Amplification of RNA Triggers via ADAR), a circuit architecture with several key advantages over existing ADAR-based translational switches. At the RNA population level, DART VADAR forms a positive feedback loop, amplifying the signal from endogenous ADAR on-demand. In addition, this topology is compact and encoded in a single transcript.

## Results

### Performance of ADAR-based riboregulators is dependent on enzyme availability

To lay the foundation for DART VADAR, we first sought to validate the hypothesis that ADAR availability is a limiting factor in RNA editing-based sensors. To do so, we designed and tested a basic ADAR-mediated sensor architecture (Supplementary Fig. 1). We used flow cytometry to quantify the expression of the mNeonGreen payload across a panel of sensors targeting different regions of a trigger encoding the iRFP720 fluorescent protein (Supplementary Fig. 2). The sensors were designed to be complementary to sequences centered on CCA sites in the trigger, with the exception of the adenosine in the stop codon. Co-transfection of plasmids encoding the sensor variants and trigger in HEK293FT cells consistently resulted in higher payload expression compared to the sensor alone, across all CCA sites tested. We observed this trend for both short (51 bp, Supplementary Fig. 3) and longer (75 bp, Fig. 1) sensor sequences; as the latter provided better performance, we proceeded with 75 bp sensor sequences for optimization of the basic ADAR-mediated sensor designs. In agreement with our hypothesis, supplying exogenous ADAR resulted in a marked increase in mNeonGreen output levels, suggesting that low endogenous levels of ADAR limit sensor performance (Fig. 1c). Of note, the p150 isoform of ADAR1 seemed to enhance output expression more than the p110 isoform, yielding up to 9-fold activation.

The output activation in cells transfected with a sensor, ADAR p150, and a trigger was readily observable at the protein level via microscopy, whereas this was not the case for cells transfected with the sensor and trigger alone (Fig. 1d). To quantify this output at the mRNA level, we evaluated the editing efficiency of sensor transcripts via next-generation sequencing (Fig. 1e). We observed over 30% editing of the adenosine in the central UAG stop codon of sensor transcripts harvested from cells transfected with a sensor, ADAR p150, and trigger. Importantly, this editing was observed to a much lesser extent

(about 3%) in cells receiving only the sensor and trigger plasmids, confirming that endogenous ADAR edits sensor transcripts but that its activity is insufficient to mediate efficient sensing. In addition, we found that ADAR-mediated editing is specific: we did not detect substantial off-target editing of other nearby adenosine residues in the sensor. These data, combined with considerations about deployment of such a technology for practical applications, prompted us to engineer sensors containing an autocatalytic feedback motif that does not require constitutive ADAR expression for sensitive detection of transcripts of interest.

We envisaged a self-amplifying circuit—DART VADAR—that consists of a sensor transcript containing four in-frame components insulated by self-cleaving 2A peptides (Fig. 1f). We cloned a transfection marker (TagBFP) upstream of the central UAG sensor to normalize for plasmid dosage. The sensor module contains a sequence complementary to a transcript of interest, with the exception of the adenosine in the central UAG stop codon. The conditionally expressed payload (the fluorescent reporter mNeonGreen) is encoded downstream of this stop codon. Further, an ADAR coding sequence is linked to the sensor output via another 2A peptide. In this system, we expected that all cells transcribing the sensors would produce TagBFP, but only cells expressing the trigger RNA would also produce mNeonGreen and exogenous ADAR.

### Identification of design rules for DART VADAR sensors

Considering the modular nature of our sensors, we first sought to independently optimize the trigger-sensor interface, starting with a simple topology in which ADAR is expressed constitutively from a separate transcript rather than conditionally from the sensor transcript. Accordingly, we set out to define general rules for the efficient targeting of RNA sequences of interest. We reasoned that since most gene-length RNA sequences harbor multiple CCA motifs, the question of which target sites should be prioritized for sensor engineering needs to be addressed. We hypothesized that the translational machinery may interfere with ADAR editing by disrupting dsRNA in coding sequences (Fig. 2a)[15]. To test this hypothesis, we compared the performance of our validated sensors targeting trigger RNA sequences in three different contexts: (a) within the original protein coding sequence; (b) in a frame-shifted construct such that the target sites are part of the mRNA 3′UTR; and (c) in the coding sequence of a secreted version of the same protein (Fig. 2a). Across all sensor sequences, targeting a secreted protein (and to a lesser extent a 3′UTR) yielded much higher activation levels than the same sites within the original protein-coding sequence—up to about 45-fold (Fig. 2b); this trend holds for other model mRNAs we tested (Supplementary Fig. 4). Since the ribosome pauses early during the translation of endoplasmic reticulum-targeted proteins[16], our observations suggested that ribosome-free RNA sequences are generally better targets. This

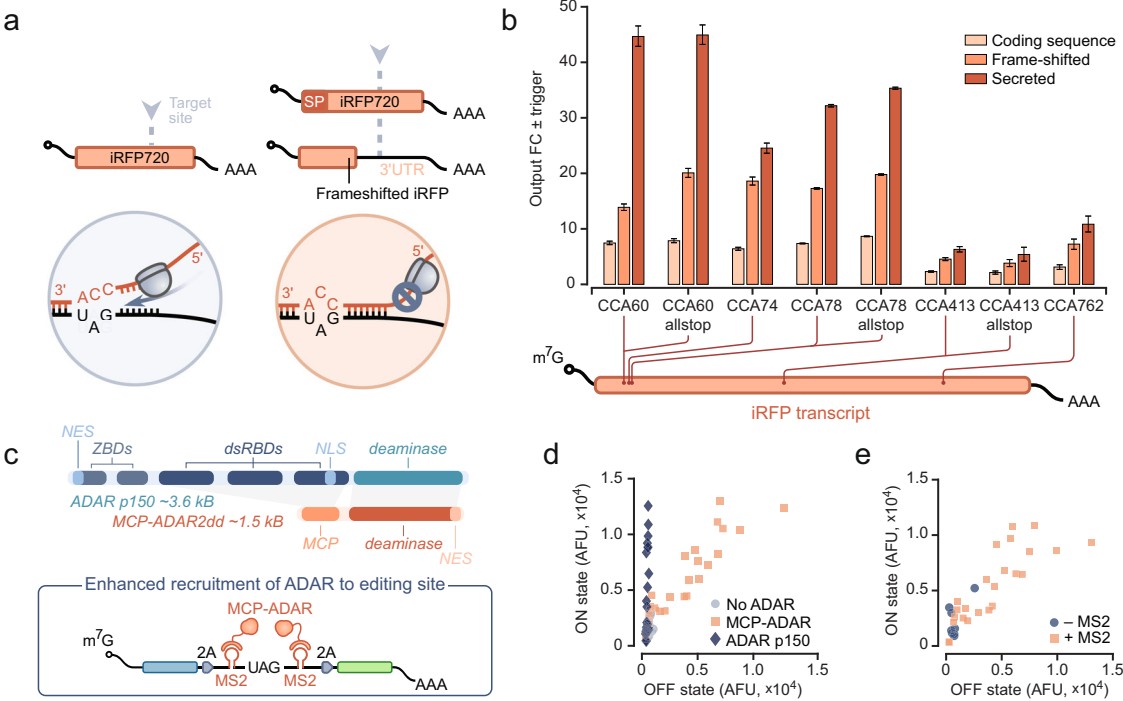

**Fig. 2 | Optimization of DART VADAR inputs and outputs. a** The performance of sensors (black) targeting a sequence in the coding sequence of a transcript is hypothesized to be diminished due to dehybridization by ribosomes translating the trigger sequence (orange). Given this, the performance of sensors designed against secreted proteins or 3'UTR sequences is expected to be enhanced as stalled and dissociated ribosomes are less likely to disrupt sensor-trigger hybridization. SP: signal peptide; UTR: untranslated region. **b** ADAR-based sensors yield higher dynamic range when designed to target the 3'UTRs of transcripts, or coding sequences of secreted proteins. "allstop" indicates that all the sites in the sensor aligning with CCA sites in the target were made into editable UAG codons (as opposed to only the central codon). The value of each bar corresponds to the output fold-change (FC), which is the ratio of the geometric mean of mNeonGreen

expression in the presence and absence of trigger. Error bars represent 95% confidence intervals for the fold-change values, determined from at least 2000 cells. **c** MCP-ADAR2dd is a compact RNA base editor, and the MCP-MS2 binding system facilitates the specific recruitment of the enzyme to the sensor edit site. NES: nuclear export sequence; ZBDs: Z-DNA binding domains; dsRBDs: dsRNA-binding domains; NLS: nuclear localization signal; MCP: MS2 major coat protein; m7G: mRNA cap; 2A: self-cleaving 2A peptide; AAA: poly(A) tail. **d** We tested the functionality of sensors containing MS2 hairpins without ADAR, with ADAR p150, or with MCP-ADAR2dd. The OFF and ON states correspond to mNeonGreen expression in the absence and presence of iRFP720 trigger mRNA, respectively. **e** MCP-ADAR specifically activates the translation of payloads encoded in sensor transcripts containing MS2 RNA hairpins.

mechanistic insight explains why preliminary descriptions of ADAR-based sensors reported highly variable performance depending on the chosen target[11]. Of note, the importance of ribosome occupancy strongly suggests that ADAR-mediated editing is cytoplasmic. To probe this further, we designed 16 sensors against the nuclear transcript MALAT1[17]. We did not observe output activation, even when we supplemented the transfection with the predominantly nuclear-localized p110 isoform of ADAR[18] (Supplementary Fig. 5). This supported and reinforced the hypothesis that in our system most editing events take place in the cytosol.

Next, we optimized the design of the modules that mediate autocatalysis in DART VADAR sensors. The natural ADAR isoforms are large (Fig. 2a), and therefore using one of these as the amplifier would undercut the delivery potential of our constructs. For instance, clinically approved adeno-associated viruses (AAVs) have a packaging limit of about 5 kilobases[19]; the coding sequence of ADAR p150 would therefore expend over 70% of that capacity. In addition, the dsRNA binding domains that mediate the recruitment of natural ADARs are promiscuous; supplying one of these ADARs in trans could therefore carry risks of off-target effects in bystander transcripts. Drawing inspiration from prior work focused on RNA-guided endogenous transcript editing[20,21], we sought to overcome these limitations by substituting natural ADARs with an engineered ADAR variant that (a) contains only the ADAR catalytic domain necessary for RNA editing, and (b) could be recruited to the edit site to increase the frequency of editing events. To this end, we used a hyperactive, minimal version of ADAR2, namely MCP-ADAR2DD(E488Q)-NES (Fig. 2c)[20]—hereafter

referred to as MCP-ADAR. The MS2 bacteriophage major coat protein (MCP) specifically binds to a short MS2 RNA hairpin and replaces the promiscuous dsRNA-interacting domains of natural ADAR enzymes with a short, localized, and orthogonal RNA-binding moiety. We integrated MCP-ADAR in-frame in the sensor transcript and added two MS2 hairpins flanking the sensor UAG codon.

Upon testing the activity of sensors modified with MS2 hairpins in human cells, we observed that the constitutive expression of MCP-ADAR results in high sensor activation in the absence of trigger (Fig. 2d), thus reducing dynamic range. However, since leaky activation by MCP-ADAR was only observed in sensors harboring MS2 hairpins (Fig. 2e), we inferred that the basal activation by MCP-ADAR is unlikely to be indicative of promiscuous activity that would result in the editing of bystander transcripts. We therefore reasoned that MCP-ADAR would be a viable option for DART VADAR if we could enhance the dynamic range by reducing background editing.

### Compact DART VADAR autocatalytic architecture boosts sensor performance

In DART VADAR sensors (Fig. 3a), MCP-ADAR is only expressed upon sensor activation. In the presence of trigger, these sensors rely on an initial editing step by endogenous ADARs, thereby yielding stop-less transcripts from which MCP-ADAR can be translated. In turn, MCP-ADAR can edit additional sensor molecules upon recruitment to the MS2 hairpins (integrated in the various ways shown in Fig. 3b), thus efficiently amplifying the initial signal. This forms a positive feedback loop in which edited sensors give rise to the enzyme that further

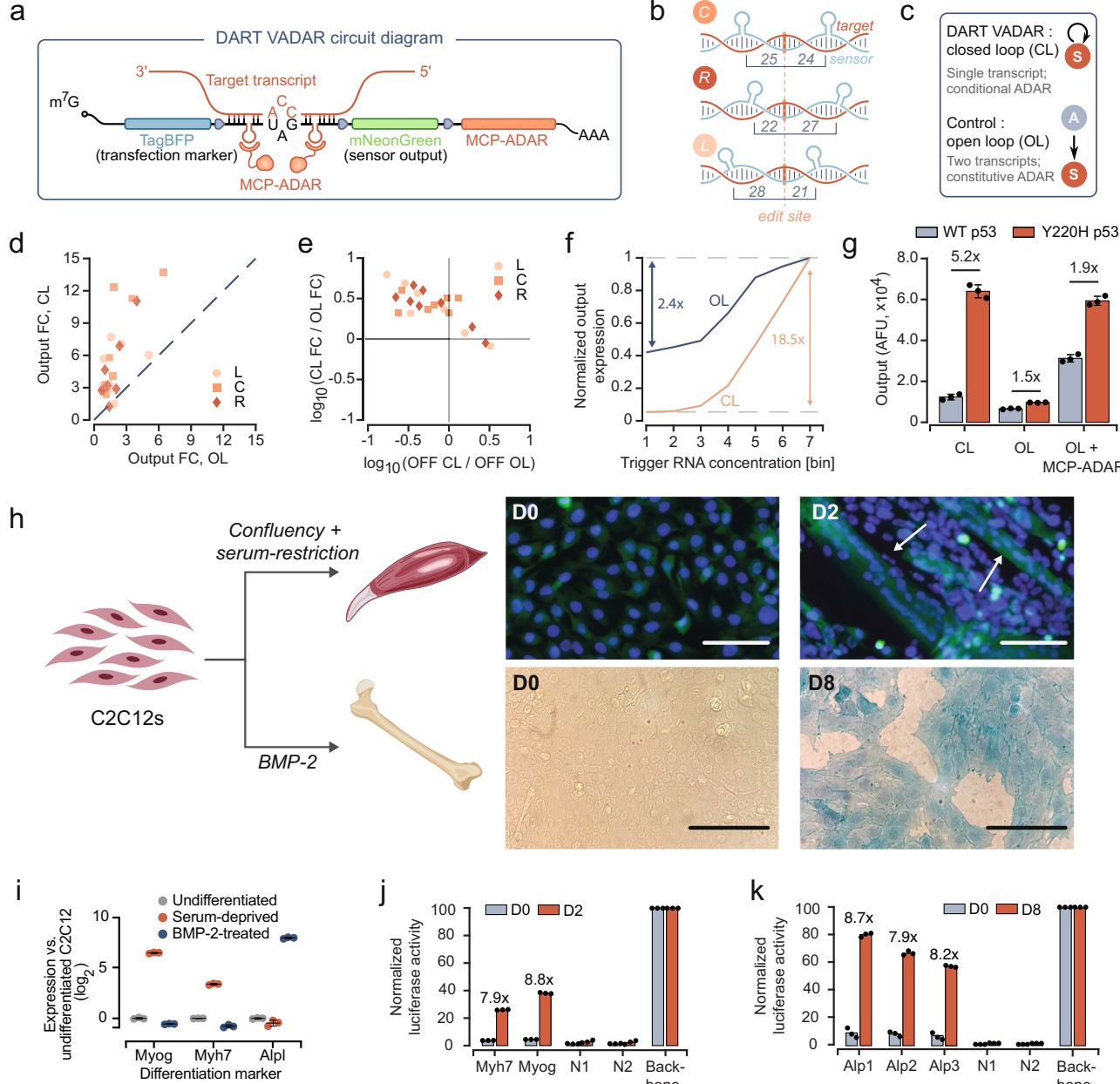

**Fig. 3 | Autocatalytic DART VADAR sensors are responsive, specific, and sensitive. a** Schematic illustrating the DART VADAR circuit design. **b** We modified ADAR-based sensors by adding flanking MS2 hairpins in three configurations. Numeric annotations correspond to the number of base pairs between the edit site and MS2 hairpin. L, left; R, right; C, center. **c** As it forms an autocatalytic loop, DART VADAR is a closed-loop (CL) system. We benchmarked DART VADAR's performance against an open-loop (OL) control, in which ADAR is constitutively expressed in trans. **d** The fold-change (FC) of the geometric mean of mNeonGreen expression levels is plotted for OL and CL variants. Points above the dashed line represent sensors that perform better with autocatalysis. **e** Closed-loop sensors with negative x-axis (basal activity ratio, CL to OL) values demonstrated a reduction of background sensor activation, and closed-loop sensors with positive y-axis (fold-change ratio, CL to OL) values showed an increase in dynamic range. **f** Open- and closed-loop sensors have different transfer curves for a given sensor expression level. The iRFP720 bin #1 corresponds to a "no-trigger" condition. For each variant, expression is normalized to the maximal sensor activation. **g** DART VADAR sensors can be designed to specifically activate in response to a point mutation ($n$ = 3 biological

replicates). WT: wild-type p53; Y220H: mutant p53. OL: open-loop, without constitutive ADAR supplementation; OL + MCP-ADAR: open-loop, with constitutive MCP-ADAR supplementation. Error bars correspond to the standard deviation for $n$ = 3 biological replicates. **h** C2C12 differentiation can be steered towards the myoblastic or the osteoblastic lineage. Top right: Hoechst and CFSE staining demonstrates the presence of multinucleated syncytia (arrows) two days post-induction of differentiation. Bottom right: alkaline phosphatase activity was detected in C2C12 cells treated with BMP-2 for 8 days. The images are representative of results from $n$ = 3 biological samples. (Scale bars: 150 μm) **i** RT-qPCR analysis highlights lineage-specific markers in undifferentiated and differentiated C2C12 cells. Bars represent mean and standard deviation measured on $n$ = 3 technical replicates. **j** Sensors targeting endogenous Myh7 and Myog mRNAs are specifically activated two days post-induction of differentiation. Backbone: stop-less sensor; N1, N2: sensors for osteoblastic differentiation. **k** Sensors targeting endogenous Alp mRNA are activated on day 8 post-treatment with BMP-2. Backbone: stop-less sensor; N1, N2: sensors for myogenic differentiation.

catalyzes this editing. We reasoned that this approach could capitalize on the low background editing by natural ADARs and the targeted and efficient editing by MCP-ADAR. In addition, the system is highly compact and can be encoded in a single transcript, potentially facilitating its delivery to cells of interest.

To benchmark the performance of DART VADAR sensors in terms of dynamic range at the protein level, we compared its activity (closed loop, CL) against an open-loop (OL) control in which MCP-ADAR is constitutively expressed in trans (Fig. 3c). We observed that, across all tested prototypes, DART VADAR yielded low background translational activation without compromising maximal activity in the presence of the trigger (Fig. 3d, e). NGS analysis of a DART VADAR sensor and its open-loop counterpart confirmed that a marked decrease in A-to-I editing in the absence of trigger underlies the reduction in background observed by flow cytometry (Supplementary Fig. 6). As a result, the great majority of tested DART VADAR prototypes had increased dynamic range compared to the open-loop system, suggesting the broad applicability of this approach for improving sensor performance (Fig. 3d, e). A poly-transfection allowed us to de-correlate the amounts of sensors and triggers[22], highlighting that the implementation of the feedback mechanism improves the transfer function for a given amount of sensor (Fig. 3f). Together, these results suggest that the DART VADAR architecture is a promising approach to generate useful in vivo sense-and-respond modules.

DART VADAR represents a relatively compact implementation of ADAR-based sensors: we found that increasing the length of the sensor sequence does not appreciably improve circuit performance in the absence of ADAR supplementation. Moreover, the use of short sensor sequences in DART VADAR RNAs avoids the undesired activation of innate dsRNA immune responses, as measured by RT-qPCR detection of the IFIH1 and IFNB1 transcripts (Supplementary Fig. 7). To further characterize the safety profile of our sensors, we quantified A-to-I editing in the transcriptome of cells expressing DART VADAR or its open-loop counterpart. The DART VADAR architecture, which features conditional expression of MCP-ADAR, reduces the frequency of off-target RNA editing compared to the constitutive expression of MCP-ADAR (Supplementary Fig. 8). Together, these results suggest that the DART VADAR architecture is a promising approach to generate useful in vivo sense-and-respond modules.

### DART VADAR sensors are specific and sensitive

To explore the utility of DART VADAR sensors for sensing cellular states, we tested their specificity and sensitivity in model mammalian cell lines. First, we investigated whether ADAR editing could be leveraged to discriminate between two RNA molecules with minimal differences. Somatic mutations are responsible for myriad complex diseases, ranging from cancer to cardiovascular and neurological conditions[23,24]. Therefore, the ability to discriminate healthy and diseased cells in mosaic tissues would be of great interest for precision therapeutics. We therefore tested whether a DART VADAR sensor targeted towards a point mutation of interest could specifically trigger translation in cells expressing a disease biomarker. As a case study, we focused on a single-base mutation in the human p53 tumor suppressor gene (c.658 T > C), which results in a Y220H substitution that is known to destabilize the DNA binding domain of p53, making it a driver of breast, lung, and liver cancers[25,26]. We transfected HEK293FT cells with a DART VADAR sensor specifically designed to detect the p53 mutant, alongside plasmids expressing either the wild-type or Y220H mutant p53 gene. The sensor was designed to be fully complementary to the Y220 codon and the surrounding sequence, such that the target adenosine could not be edited by ADAR; conversely, imperfect hybridization with the mutant mRNA produced a single-base-pair bulge, exposing the adenosine for editing by ADAR. We observed a fivefold activation in the reporter gene downstream of the sensor in cells expressing p53-Y220H, highlighting the specificity of DART VADAR sensors (Fig. 3g).

We next investigated whether DART VADAR could be used to discriminate closely related cell types based on deferentially expressed endogenous genes. We focused our study on the C2C12 murine myoblast cell line, a well-described model of cell differentiation (Fig. 3h). When they reach confluency, and particularly in serum-restricted conditions, C2C12 cells differentiate to form functional myotubes. Alternatively, upon exposure to bone morphogenetic protein-2 (BMP-2), the cells are biased to differentiate towards an osteoblastic lineage[27]. We designed sensors targeting RNA markers of both cell fates, namely the 3' UTRs of mRNAs encoding myogenin and slow-twitch myosin heavy chain I (two proteins expressed during myogenesis), and the coding sequence of alkaline phosphatase (a bone-mineralizing enzyme). We then differentiated C2C12 cells, which we confirmed phenotypically either by the presence of multinucleated syncytia indicative of early myotube formation (eventually forming functional contractile units, presented in Supplementary Movie S1), or by the detection of strong alkaline phosphatase activity (Fig. 3h). Reverse transcription followed by quantitative PCR (RT-qPCR) also confirmed the expected increase in the expression of the target mRNAs (Fig. 3i), indicative of the transcriptional changes that drive differentiation. We observed that our DART VADAR constructs expressed their payload (Nanoluc luciferase) as a response: sensors targeting the myogenin and myosin mRNAs were activated in myotubes (Fig. 3j), while alkaline phosphatase-targeting sensors were strongly activated in BMP-2-induced osteoblasts (Fig. 3k)—up to 80% of the maximum level defined by a stop-less control. These observations demonstrate that DART VADAR constructs are sensitive enough to drive high levels of expression of user-defined payloads in response to endogenous levels of natural transcripts, making them well suited to sense and respond to transcriptional changes across both cell types and cell states.

## Discussion

In this work, we presented DART VADAR, a sensitive, programmable, modular, and compact RNA sense-and-respond circuit. Hybridization of a DART VADAR sensor with a user-defined trigger transcript initiates RNA editing of a premature stop codon, driving the translation of the downstream payload sequences. We validated a secondary payload in the form of a hyperactive, minimal version of ADAR2 and targeted it to the edit site via the MS2 RNA hairpin-coat protein interaction, resulting in an autocatalytic positive feedback loop. This configuration relies on endogenous ADAR to elicit the initial response with a high degree of specificity. We demonstrated that by using autocatalysis, we attenuated the circuit background and enhanced the output dynamic range by close to eightfold relative to an open-loop configuration, while reducing the overall number of components and genetic footprint of the technology. The resulting circuit is able to detect minimal differences between RNA molecules and interpret endogenous signals to control transgene expression across different cell states.

While we have defined general rules for targeting user-defined RNA targets, the choice of a target site for in vivo applications might involve additional considerations beyond editing efficiency. Machine learning would be a straightforward way to optimize target detection, as has been done for toehold switches[28,29], but the design of editing-based sensors might involve unique trade-offs between efficacy and safety. Sensor sequences encode translated peptides, the exact sequences of which are defined by the target RNA sites; different sensors targeting a given RNA sequence will therefore produce different peptides, which might vary in their immunogenicity. We expect that recent computational advances in the prediction of peptide immunogenicity could be leveraged to further refine the prediction of optimal target sites[30], thereby guiding the design of therapeutically relevant DART VADAR sensors.

Our work expands the application space of editing-based riboregulators: the autocatalytic feedback implementation features

a size of less than 5 kb (including promoter and terminator), making it amenable for delivery in clinically relevant vectors[19]. Importantly, as ADAR enzymes are endogenously expressed in most human tissues[14], we expect most cells to be able to trigger autocatalysis when provided with DART VADAR sensors. We envision that DART VADAR could lay the foundation for easy-to-deliver smart RNA-based therapeutics.

## Methods

### Cloning

For the sensor expression plasmids, we built custom entry vectors by isothermal assembly of dsDNA fragments using the NEBuilder HiFi DNA assembly mix (NEB #E2621). We generated fragments by PCR using high-fidelity Q5 polymerase (NEB #M0494), with in-house plasmids and custom-synthesized gBlocks (Integrated DNA Technologies) as templates. We designed the entry vectors such that the fluorescent protein expression cassettes harbor a multiple cloning site (MCS) without in-frame stop codons, insulated from the fluorescent proteins by sequences coding for 2A peptides. To assemble the final sensor plasmids, we ordered sensor sequences as long oligonucleotides (Sigma Aldrich) with 5' and 3' adapter sequences overlapping with the vectors around the HindIII site of the MCS. We made the oligonucleotides double-stranded by PCR and inserted the resulting dsDNA products into HindIII-linearized entry vectors using HiFi assembly mix. To build ADAR-expressing plasmids, we used plasmids pmGFP-ADAR1-p150 and pmGFP-ADAR1-p110, kindly provided by Kumiko Ui-Tei (Addgene #117927 and #117928, respectively) as a starting point. We excised the GFP sequences from the plasmids by amplifying the backbones with Q5 polymerase before circularizing the PCR products with KLD mix (NEB #M0554). The MCP-ADAR sequence was amplified from plasmid MS2-adRNA-MCP-ADAR2DD(E488Q)-NES, kindly provided by Prashant Mali (Addgene #124705). After each cloning and transformation step, we verified the regions of interest in individual clones by Sanger sequencing (QuintaraBio, Azenta). We propagated all the plasmids in *Escherichia coli* Turbo (NEB #C2984) or Stable (NEB #C3040) strains, with 100 µg/mL carbenicillin (Teknova #C2110) for selection.

### Human cell culture

We obtained cryopreserved HEK293FT cells from Invitrogen (#R70007) and maintained them in Dulbecco's modification of Eagle's medium (DMEM, Gibco #10569010) supplemented with 10% v/v fetal bovine serum (FBS, Gibco #16000044) and 1X MEM non-essential amino acids (Gibco #11140050). Both wild-type and MALAT1 knock-out A459 cells were kindly provided by Sven Diederichs (DKFZ Heidelberg, Germany)[31]; we propagated these cells in Ham's F-12K (Kaighn's) Medium (Gibco #21127030) supplemented with 10% v/v FBS. We grew all the cells in a humidified atmosphere at 37 °C with 5% $CO_2$, and split the cells using trypsin-EDTA (Gibco #25300054) every 2–3 days to ensure they did not surpass 80% confluence. We used cells at low passage numbers (<15) for all experiments.

### C2C12 cell culture and differentiation

We obtained C2C12 cells from ATCC (CRL-1772) and maintained the cells in culture in DMEM supplemented with 10% v/v FBS. Care was taken to ensure that they did not exceed 50% confluence. For differentiation to the muscle lineage, we allowed the cells to become fully confluent (which we defined as day 0) one day after transfections, at which point we switched the growth medium to DMEM supplemented with 2% v/v horse serum (Cytiva #SH3007402) and 1X insulin-transferrin-selenium supplement (Sigma Aldrich #I3146). We replaced the growth medium every 48 hr until the end of the differentiation experiment. For differentiation to the bone lineage, we grew the cells in DMEM + 10% v/v FBS supplemented with 1000 ng/mL

recombinant BMP-2 (R&D Systems #355BEC025) for 5 days prior to transfection with DART VADAR sensor plasmids.

### Transfections

We used Lipofectamine 3000 (Invitrogen #L3000015) for transient transfections. We transfected cells at 70–90% confluence. In each well of a 96-well plate, we transfected a total of 150 ng plasmid DNA, which included 50 ng of each plasmid (sensor, ADAR, and/or target). When leaving out one or several plasmids, we standardized the mass of transfected plasmids by adding a filler plasmid (carrying an Fluc2 gene with or without a promoter). For each well, we diluted 0.5 µL of P3000 reagent in a final volume of 5 µL of OptiMEM (Gibco #51985091), as well as 0.5 µL of Lipofectamine in 5 µL of OptiMEM. For larger culture vessels, we scaled up the transfections according to the area of the plates. We analyzed the cells 48 h after transfection.

### Fluorescence analyses

We analyzed fluorescent protein expression by flow cytometry. To do so, we harvested cells 48 h after transfection using trypsin-EDTA. We washed the cells three times with flow cytometry buffer, made of phosphate-buffered saline without calcium or magnesium (Corning #21031CV) supplemented with 1% FBS and 5 mM EDTA. We kept cells on ice until analysis with the HTS module of a BD LSR-II flow cytometer (Koch Institute flow cytometry core). We determined voltage settings for each relevant channel using BD FACSDiva. We analyzed the data using Matlab scripts (based on https://github.com/jonesr18/MATLAB_Flow_Analysis). As a general strategy, we binned cell populations according to their transfection levels, at half-log intervals in the TagBFP-Pacific Blue channel (Supplementary Fig. 2).

### Microscopy

For the imaging of HEK293FT cells, we transfected cells in a tissue-culture treated polystyrene 24-well plate. After 48 h, we replaced the growth medium of the transfected cells with Hank's balanced salts solution without phenol red (Sigma Aldrich #H6648) and proceeded with the imaging at room temperature. We collected the images on a Nikon Ti-E inverted microscope equipped with a Nikon CFI S Plan Fluor ELWD 20 × 0.45 NA objective. We used a Nikon Intensilight C-HGFIE mercury lamp for illumination, and the following filters for mNeonGreen: a 470/40 excitation filter and a 425/50 emission filter (Chroma #49002). We acquired images with a Hamamatsu ORCA-Flash 4.0 CMOS camera controlled with NIS Elements AR 4.13.05 software.

### RNA editing analysis

We transfected HEK293FT cells in triplicates in 6-well plates with the appropriate plasmids. After 48 h, we harvested the cells with trypsin-EDTA; we then washed the cells with flow cytometry buffer and used fluorescence-activated cell sorting (FACS) to isolate transfected cells (TagBFP-positive cells, detected in the Pacific Blue channel). We sorted the cells directly in the lysis buffer from the Qiagen RNeasy Mini kit (#74106), and stored the homogenized samples at −80 °C until we proceeded with total RNA extraction following the manufacturer's instructions. A third-party company (Quintara Biosciences) produced cDNAs by reverse transcription of the sensor regions using an Easy-Quick RT MasterMix (Cwbio #CW2019M) and primers CCA60-2264F/CCA60-2463R described in Supplementary Table 1, after which the samples were sequenced on an Illumina MiSeq platform using a MiSeq Reagent Nano Kit v2 (300-cycles). We estimated editing efficiency by aligning the reads of each sample using Geneious mapper at medium sensitivity with up to 5 iterations per alignment, and used a custom Matlab script to detect A-to-G substitutions at each nucleotide position.

## Off-target editing analysis

We prepared samples for RNA sequencing using a Qiagen RNeasy Plus mini kit (#74136). A third-party company (Quintara Biosciences) enriched the mRNAs using NEBNext Poly(A) mRNA Magnetic Isolation Module, followed by workup with NEBNext UltraII Directional RNA Library Prep Kit for Illumina. Sequences were sequenced on an Illumina MiSeq platform using a MiSeq Reagent Micro Kit v2 (2×150-cycles). We trimmed the reads using Fastp[32], indexed them, and used STAR[33] to align the reads to the UCSC hg38 reference genome[34] and annotations from Gencode[35]. We analyzed the sorted BAM files for A-to-I edits using REDItools v1.3[36,37]. The parameters can be found online at https://github.com/joncchen/dart_vadar.

## Quantification of gene expression

At each timepoint, we harvested cells from 16 wells of a 96-well plate, or a single well of a 6-well plate, depending on the experiment. We used a total of 1 mL of Tri-reagent RT (Molecular Research Center #RT111), and vortexed the samples vigorously for 5 min. We stored samples in the Tri-reagent at −80 °C until extraction. After thawing the samples, we added 50 μL of 4-bromoanisole (Thermo Scientific #A1182422) to the homogenate, vortexed, and stored the samples on ice for 5 min prior to a 21,000 × $g$ centrifugation at 4 °C for 15 min. From the upper, clear aqueous phase, we carefully harvested 500 μL that we thoroughly mixed 1:1 with isopropanol. We applied the mixture on a silica spin column (Epoch Life Science #1910), which we spun and washed three times, once with buffer RW1 (Qiagen #1053394) and twice with buffer RPE (Qiagen #1018013). We eluted the samples in nuclease-free water and checked the RNA quality and concentration using a Nanodrop spectrophotometer. We then used about 100 ng of total RNA in each RT-qPCR reaction using the Luna Universal One-Step RT-qPCR Kit (NEB #E3005), set up according to manufacturer's instructions and analyzed on a CFX Opus 96 instrument (Bio-Rad) using Bio-Rad CFX Maestro 2.0 in the SYBR-green channel. We normalized C2C12 gene expression within each biological sample to the levels of the housekeeping gene Csnk2a2. Primer sequences are available in Supplementary Table 1.

## Luciferase assays

At each time point of interest, we sacrificed wells transfected with the combinations of plasmids of interest. In each well of a 96-well plate containing 100 μL of growth medium, we added another 100 μL of Nano-Glo lysis/reaction buffer (Promega #N1110/N3040) reconstituted following manufacturer's recommendations. We vigorously pipetted to ensure complete homogeneization, and incubated the samples for 5 min at room temperature; we then transferred 150 μL of each sample to a white-bottom 96-well plate and measured luminescence on a ClarioStar Plus instrument (BMG Labtech, 5.70 R2) set with an acquisition window of 480/70 nm.

## Staining

We stained C2C12 cells with Hoechst 33342 (Thermo Fisher #62249) and carboxyfluorescein succinimidyl ester (CFSE, Invitrogen #65085084) during differentiation to the muscle lineage. We diluted 1 μL Hoechst staining solution (20 mM stock) and 1 μL of 1000x CFSE (10 mM stock in DMSO) in 1 mL PBS and added 100 μL of this working staining solution to one well in a 96-well plate. We incubated the samples at room temperature protected from light for 10 min. Afterwards, we washed the wells three times with PBS prior to imaging on an EVOS M5000 microscope equipped with DAPI and GFP light cubes (Invitrogen #AMEP4950, AMEP4951). To generate images overlaying the DAPI (Hoechst) and GFP (CFSE) channels, we used the "Merge channels" function in Fiji 2. For the functional evaluation of alkaline phosphatase expression in C2C12s treated with BMP-2, we fixed the cells with a paraformaldehyde-based buffer (Biolegend #420801) for 10 min at room temperature protected from light. We then washed the cells with water and subsequently stained with nitro blue tetrazolium chloride and 5-bromo-4-chloro-3-indolyl-1-phosphate (BCIP/NBT, Sigma Aldrich #AB0300). We then washed the samples again with water, prior to imaging with an iPhone 12 mini (dual 12 MP, f/1.6 aperture, and iOS 15.5 software) mounted on a light transmission microscope.

## Statistics and reproducibility

The sample size for each subpopulation was imposed by the number of cells growing in culture vessels. In general, the presented summary statistic metrics were calculated on binned cell subpopulations, which contained at least 3000 cells; this number of cells ensures that the calculated metrics are not biased by noise. For many of the experiments, we chose a sample size of $n = 3$ (three experimental repeats) because it is standard practice in the field. No statistical method was used to pre-determine sample size. No data were excluded from the analyses, with the exception of flow cytometry data which we gated following the rationale described in Supplementary Fig. 2. We performed all key experiments at least twice; results were consistent across these replicates and the data presented in the article are representative of the trends we observed. The experiments were not randomized as we performed all the experiments summarized in the manuscript using immortalized cell lines, which can reasonably be assumed to be identical when split into parallel wells for transfection. The investigators were not blinded to allocation during experiments and outcome assessment as we pre-defined metrics of success for the characterization of DART VADAR performance versus analogous open-loop controls (i.e., fold-changes, background activation).

## DART VADAR sensor design

The general workflow for designing DART VADAR sensors is described in Supplementary Fig. 9.

## Reporting summary

Further information on research design is available in the Nature Portfolio Reporting Summary linked to this article.

# Data availability

All data needed to evaluate the conclusions in the study can be found in the paper and/or the supplementary materials. Source data accompanying this manuscript include measured fold-changes, editing rates, and RT-qPCR calculations. New plasmids used in this study are available for distribution from Addgene. Source data are provided with this paper. RNA sequencing data have been deposited in the Sequence Read Archive (SRA) database in the BioProject PRJNA932010. Raw.fcs files and other data are available from the corresponding author. Plasmids are available to the scientific community through Addgene. Correspondence and requests for materials should be addressed to J.J.C. Source data are provided with this paper.

# Code availability

General MATLAB code for use in.fcs file processing and analysis are available under an open-source license in the GitHub repository https://github.com/jonesr18/MATLAB_Flow_Analysis. Specific.m scripts for each experiment are available from the corresponding authors upon reasonable request. The parameters used in the code for the RNA-sequencing data analysis can be found online at https://github.com/joncchen/dart_vadar.

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

## Acknowledgements

We are indebted to Domitilla Del Vecchio for her support. We are grateful to Kumiko Ui-Tei and Prashant Mali for providing ADAR plasmids and to Sven Diederichs for providing wild-type and MALAT1-KO A549 cells. We thank the flow cytometry core at the Koch Institute's Robert A. Swanson (1969) Biotechnology Center for technical support. We thank all the members of the Collins laboratory for their advice and help throughout the course of this study, and Max A. English for helpful comments on the manuscript. We also thank Lin-Ya Huang for technical advice and Benjamin Chang for help with RNA-seq annotations. This work was funded by NIH grants R01EB024591 (J.J.C.) and 5RC2DK120535-03 (J.J.C.), and the Wyss Institute for Biologically Inspired Engineering.

## Author contributions

Conceptualization: R.V.G., K.I., S.R., N.D.T., and J.J.C.; methodology: R.V.G., K.I., S.R., and N.D.T.; investigation: R.V.G., K.I., S.R., N.D.T., M.A.L., K.Z., J.X.C., and J.C.C.; validation: R.V.G., K.I., S.R., N.D.T., M.A.L., K.Z., J.X.C., and J.C.C.; formal analysis: R.V.G. and K.I.; visualization: R.V.G. and K.I.; software: R.V.G., K.I., N.D.T., K.Z., J.C.C., and J.V.A.; writing—original draft: R.V.G., K.I., S.R., N.D.T., M.A.L., K.Z., J.X.C., and J.C.C.; writing—review and editing: all authors; supervision and funding acquisition: J.J.C.

## Competing interests

J.J.C. is an inventor on multiple patents that cover RNA riboregulators. He is a cofounder of Synlogic and Senti Biosciences, a cofounder and director of Sherlock Biosciences, and is on the Shape Therapeutics scientific advisory board. R.V.G., K.I., S.R., N.D.T., M.A.L., K.Z., J.X.C., J.C.C., and J.J.C. have filed a provisional patent (with Massachusetts Institute of Technology as the applicant) on covering the DART VADAR circuitry described herein (US Provisional Application Number 63/481,010). The remaining authors declare no competing interests.
