## [Peer Review File · Nature Communications]

Reviewers' Comments:

Reviewer #1:

Remarks to the Author:

In this manuscript, the authors design a programmable RNA sensor that converts target hybridization into a translational output. Recently, several studies have developed similar systems to detect the expression of endogenous transcripts through an RNA sensor. In all these systems, RNA base editing by ADARs couples the detection of an RNA trigger to the translation of a user-defined genetic payload. Compared to other systems, the authors amplify the signal from editing by endogenous ADAR through a positive feedback loop. Amplification is mediated by the expression of a hyperactive ADAR-MCP fusion protein and its recruitment to the edit site via MS2 hairpin structure.

Overall, this study develops a novel RNA sensor system, and this system can be used to amplify the low level of RNA base editing via the positive feedback loop.

Major points

1. The authors only tested 51bp and 75bp sensors and demonstrated the advantage of VADAR. Since longer sensors may lead to higher editing levels, there may be no need to amplify the signal via VADAR. Thus, the authors need to demonstrate that the VADAR system has a better performance for sensors with various lengths (e.g. 100bp, 150bp, 250bp ...).
2. The overexpression of ADAR-MCP function may lead to a global off-target effect, which greatly limits the application of such a system, particularly in therapeutic settings. Thus, the authors need to perform RNA-seq to evaluate the global off-target effects of their system and make additional efforts to optimize the system to minimize the off-target effect.
3. The manuscript seems in a rush and there are not enough details provided in the methods for the work to be reproduced. e.g. figure 3g. The authors said that they transfected HEK293FT cells with a DART VADAR sensor specifically designed to detect the p53 mutant. However, no details are described on how they designed a sensor that can distinguish a single mutation?
4. For most of the figures, the authors need to quantify the editing efficiency of the sensors directly but not just use the fluorescence intensity.

Minor points:

1. Since the authors suggest that ribosome-free proportions of mRNAs are better targeted, the conclusion will be more convincing if more targets are tested.

Reviewer #2:

Remarks to the Author:

Gayet, R.V., et. al.

The article describes an improvement in the space of RNA-based sensors using adenosine deaminases acting on RNA (ADARs). Recently, ADARs have been used as RNA-based sensors that can be coupled with translation of a genetic payload using specific hybridization and the ADAR base editing capabilities. However, previous technology falls short in cells with low levels of endogenous ADARs, and constitutive of exogenous ADARs insufficiently solve this issue, since this results in increased transcriptional units and potential off-target effects. This study proposes an autocatalytic model of ADAR sensors, in which the sensor is activated by endogenous ADAR but is then amplified by its own ADAR payload, resulting in higher dynamic range, lower background, and fewer off-target effects.

The article is clearly written and was a pleasure to read. The data supports the conclusions (I have one comment below). The results are very important to the field of synthetic biology, in particular in the area of targeted drug delivery in the context of microbe-based therapies. I highly

recommend publication in Nat. Comm.

For the last figure, I'd like to see a control experiment with constitutively expressed ADAR, along with the DART VADAR sensor in sensing the WT v. mutant p53 gene.

I thank the authors for a very clear manuscript that was quite easy to evaluate.

Manuscript reference: NCOMMS-22-26413

Title: Autocatalytic base editing for RNA-responsive translational control

Corresponding author: Prof. James J. Collins (MIT)

Response to reviewers

We would like to thank the reviewers for their constructive comments and suggestions. We have revised the paper to address each of their points, and we think these changes have helped strengthen the paper considerably. Below, we provide responses to each point raised by the reviewers, along with descriptions of the associated revisions introduced to the paper. Our responses are highlighted in orange.

Reviewer #1

In this manuscript, the authors design a programmable RNA sensor that converts target hybridization into a translational output. Recently, several studies have developed similar systems to detect the expression of endogenous transcripts through an RNA sensor. In all these systems, RNA base editing by ADARs couples the detection of an RNA trigger to the translation of a user-defined genetic payload. Compared to other systems, the authors amplify the signal from editing by endogenous ADAR through a positive feedback loop. Amplification is mediated by the expression of a hyperactive ADAR-MCP fusion protein and its recruitment to the edit site via MS2 hairpin structure.

Overall, this study develops a novel RNA sensor system, and this system can be used to amplify the low level of RNA base editing via the positive feedback loop.

Major point 1. The authors only tested 51bp and 75bp sensors and demonstrated the advantage of VADAR. Since longer sensors may lead to higher editing levels, there may be no need to amplify the signal via VADAR. Thus, the authors need to demonstrate that the VADAR system has a better performance for sensors with various lengths (e.g. 100bp, 150bp, 250bp . . .).

The reviewer raises an important point: intuitively, longer sensor sequences allow for the formation of longer dsRNA helices. These may be more stable and more likely to be edited, thereby improving the dynamic range in the absence of exogenous ADAR. To test this hypothesis, we set up an additional experiment to compare the performance of sensors regulated either by endogenous ADAR, or through the autocatalytic mechanism of DART VADAR. The results, shown in Supplementary Fig. S7A, demonstrate that increasing the length of the sensor sequence does not appreciably improve circuit performance in the absence of ADAR supplementation. Moreover, we observed that increasing the length of sensor-target duplexes results in a noticeable activation of the innate immune response, as measured by RT-qPCR detection of the IFIH1 and IFNB1 transcripts (Supplementary Fig. S7B,C). The DART VADAR technology allows us to use shorter sensor sequences, thus minimizing such adverse cellular responses. We have revised the paper to include this new work.

Major point 2. The overexpression of ADAR-MCP function may lead to a global off-target effect, which greatly limits the application of such a system, particularly in therapeutic settings. Thus, the authors need to perform RNA-seq to evaluate the global off-target effects of their system and make additional efforts to optimize the system to minimize the off-target effect.

We agree that assessing the off-target RNA editing rate is crucial in evaluating the applicability of our technology in clinical contexts. In our initial submission, the comparison of sensors with and without ADAR-recruiting MS2 RNA hairpins demonstrated that sensor RNA molecules were not edited if they were not specifically designed to interact with the base editor. To confirm the specificity of editing, we transfected HEK293FT cells with several sensor variants (i.e., DART VADAR and an open-loop counterpart with no ADAR, MCP-ADAR, and ADAR p150) and performed transcriptome-wide RNA sequencing to look for global off-target effects. The results, shown in Supplementary Figure S8, indicate that A-to-I off-target edits in cells expressing DART VADAR are less frequent than in similar conditions with their open-loop counterparts. These data suggest that the autocatalytic mechanism, which regulates MCP-ADAR expression, mitigates concerns about potential off-target effects associated with using a hyperactive base editor. We have revised the paper to include this new work.

Major point 3. The manuscript seems in a rush and there are not enough details provided in the methods for the work to be reproduced. e.g. figure 3g. The authors said that they transfected HEK293FT cells with a DART VADAR sensor specifically designed to detect the p53 mutant. However, no details are described on how they designed a sensor that can distinguish a single mutation?

We agree with the reviewer on the importance of explaining how we designed our sensors. We have included additional text in the revised manuscript to guide the readers (copied below).

"We transfected HEK293FT cells with a DART VADAR sensor specifically designed to detect the p53 mutant, alongside plasmids expressing either the wild-type or Y220H mutant p53 gene. *The sensor was designed to be fully complementary to the Y220 codon and the surrounding sequence, such that the target adenosine could not be edited by ADAR; conversely, hybridization with the mutant mRNA produced a bulge, exposing the adenosine for editing by ADAR.* We observed a 5-fold activation in the reporter gene downstream of the sensor in cells expressing p53-Y220H, highlighting the specificity of DART VADAR sensors"

Major point 4. For most of the figures, the authors need to quantify the editing efficiency of the sensors directly but not just use the fluorescence intensity.

The level of RNA editing is indeed important for gaining a mechanistic insight into the behavior of ADAR-based sensors. We performed an additional NGS analysis of open-loop and closed-loop RNA sensors (Supplementary Fig. S6), confirming that a marked increase in A-to-I editing underlies the translational activation that we measured by flow cytometry. Interestingly, we found that while A-to-I editing efficiency is highly correlated with output mNeonGreen expression levels for a given circuit architecture, this is not the case when comparing across different types of ADAR-based sensors (i.e., DART VADAR vs. constitutive overexpression of ADAR). These results highlight that translational readouts are more informative because they

integrate several important layers of regulation: they reflect not only RNA editing levels, but also related biological processes. For instance, it is well-established that the length, sequence, and ribosomal occupancy of mRNAs are some determinants of transcript stability. All these factors vary across experimental conditions and sensor architectures. We therefore reasoned that the protein output (e.g., the therapeutic payload in a clinical setting) is the most relevant metric in the context of real-life applications.

Minor point 1. Since the authors suggest that ribosome-free proportions of mRNAs are better targeted, the conclusion will be more convincing if more targets are tested.

We agree with the reviewer's suggestion and have tested our hypothesis on additional model targets. We created sensors against mRNAs encoding NanoLuc luciferase or puromycin acetyltransferase (Supplementary Fig. 4). We found that across all 16 prototypes, sensors activated by either a frame-shifted or secreted trigger consistently yielded the highest dynamic ranges. These results are in line with our previous observations and support the hypothesis that ribosomal occupancy affects sensor performance. We have revised the paper to include this new work.

Reviewer #2

The article describes an improvement in the space of RNA-based sensors using adenosine deaminases acting on RNA (ADARs). Recently, ADARs have been used as RNA-based sensors that can be coupled with translation of a genetic payload using specific hybridization and the ADAR base editing capabilities. However, previous technology falls short in cells with low levels of endogenous ADARs, and constitutive of exogenous ADARs insufficiently solve this issue, since this results in increased transcriptional units and potential off-target effects. This study proposes an autocatalytic model of ADAR sensors, in which the sensor is activated by endogenous ADAR but is then amplified by its own ADAR payload, resulting in higher dynamic range, lower background, and fewer off-target effects.

The article is clearly written and a was a pleasure to read. The data supports the conclusions (I have one comment below). The results are very important to the field of synthetic biology, in particular in the area of targeted drug delivery in the context of microbe-based therapies. I highly recommend publication in Nat. Comm.

For the last figure, I'd like to see a control experiment with constitutively expressed ADAR, along with the DART VADAR sensor in sensing the WT v. mutant p53 gene.

We thank the reviewer for this suggestion. We repeated the p53 mRNA sensing experiment, comparing open-loop (control) and closed-loop (DART VADAR) sensor architectures. The results, presented in Figure 3G, confirm that autocatalytic amplification indeed improves the performance of RNA sensors in the context of single-base discrimination. We have revised the paper to include this new work.

I thank the authors for a very clear manuscript that was quite easy to evaluate.

Reviewers' Comments:

Reviewer #1:

Remarks to the Author:

The authors addressed most of my concerns. I have only one comment remaining.

Fig S8. The current plots don't show the number of global off-targets, as well as the levels of changes of these off-targets. I suggest the authors use scatter plots to perform pairwise comparisons to show editing levels of all detected sites in cells transfected with different sensor variants.

Reviewer #2:

Remarks to the Author:

The authors have done an outstanding job addressing my comments - I highly recommend publication.

Manuscript reference: NCOMMS-22-26413

Title: Autocatalytic base editing for RNA-responsive translational control

Corresponding author: Prof. James J. Collins (MIT)

Response to reviewers

We would like to thank the reviewers for their additional constructive comments and suggestions. We have revised the paper to address these points, and we think these changes have helped strengthen the paper. Below, we provide responses to each point raised by the reviewers, along with descriptions of the associated revisions introduced to the paper. Our responses are highlighted in orange.

Reviewer #1

The authors addressed most of my concerns. I have only one comment remaining.

Fig S8. The current plots don't show the number of global off-targets, as well as the levels of changes of these off-targets. I suggest the authors use scatter plots to perform pairwise comparisons to show editing levels of all detected sites in cells transfected with different sensor variants.

We thank the reviewer for their suggestion, and we believe it led us to better present the results of the RNA sequencing experiment. We have added the suggested pairwise scatter plots and a bar chart quantifying the frequency of off-target effects in each sample.

Reviewer #2

The authors have done an outstanding job addressing my comments - I highly recommend publication.

We thank the reviewer again for their previous comments.